**Data Availability Statement:** Data from the Alzheimer's Disease Neuroimaging Initiative (ADNI) may be requested through the ADNI website (http://adni.loni.usc.edu/data-samples/access-data/

# Cognition and motor function: The gait and cognition pooled index

Jacqueline K. Kueper[1], Daniel J. Lizotte[1,2,3,4], Manuel Montero-Odasso[1,5]*, Mark Speechley[1,4], for the Alzheimer's Disease Neuroimaging Initiative[¶]

**1** Department of Epidemiology & Biostatistics, Schulich School of Medicine & Dentistry, University of Western Ontario, London, Ontario, Canada, **2** Department of Computer Science, Faculty of Science, University of Western Ontario, London, Ontario, Canada, **3** Department of Statistical and Actuarial Sciences, Faculty of Science, University of Western Ontario, London, Ontario, Canada, **4** Master of Public Health Program, Schulich School of Medicine & Dentistry, University of Western Ontario, London, Ontario, Canada, **5** Division of Geriatric Medicine, Schulich School of Medicine & Dentistry, University of Western Ontario, London, Ontario, Canada

¶ Data used in preparation of this article were obtained from the Alzheimer's Disease Neuroimaging Initiative (ADNI) database (adni.loni.usc.edu). As such, the investigators within the ADNI contributed to the design and implementation of ADNI and/or provided data but did not participate in analysis or writing of this report. A complete listing of ADNI investigators can be found at: http://adni.loni.usc.edu/wp-content/uploads/how_to_apply/ADNI_Acknowledgement_List.pdf

* mmontero@uwo.ca

## Abstract

### Background

There is a need for outcome measures with improved responsiveness to changes in pre-dementia populations. Both cognitive and motor function play important roles in neurodegeneration; motor function decline is detectable at early stages of cognitive decline. This proof of principle study used a Pooled Index approach to evaluate improved responsiveness of the predominant outcome measure (ADAS-Cog: Alzheimer's Disease Assessment Scale-Cognitive Subscale) when assessment of motor function is added.

### Methods

Candidate Pooled Index variables were selected based on theoretical importance and pairwise correlation coefficients. Kruskal-Wallis and Mann-Whitney U tests assessed baseline discrimination. Standardized response means assessed responsiveness to longitudinal change.

### Results

Final selected variables for the Pooled Index include gait velocity, dual-task cost of gait velocity, and an ADAS-Cog-Proxy (statistical approximation of the ADAS-Cog using similar cognitive tests). The Pooled Index and ADAS-Cog-Proxy scores had similar ability to discriminate between pre-dementia syndromes. The Pooled Index demonstrated trends of similar or greater responsiveness to longitudinal decline than ADAS-Cog-Proxy scores.

). Data from the Gait and Brain study cannot be shared publicly because the study is still ongoing and the data contain personal health information. This restriction has been imposed by the ethics board of the University of Western Ontario. Data access queries can be directed to Yanina Sarquis-Adamson (contact via Yanina. SarquisAdamson@sjhc.london.on.ca). The authors of the present study had no special access privileges in accessing the underlying data which other researchers would have.

**Funding:** This work was supported by the Early Research Award (PI MMO) and the Alzheimer Foundation of London and Middlesex Master's Scholarship in Alzheimer-Related Research to JKK. The Gait and Brain Study is funded by an operating grant from the Canadian Institutes of Health and Research (CIHR, MOP 211220) and a CIHR Project Grant (PJT 153100). Dr. Montero-Odasso's program in "Gait and Brain Health" is supported by grants from the CIHR, the Ontario Ministry of Research and Innovation, the Ontario Neurodegenerative Diseases Research Initiative, the Canadian Consortium on Neurodegeneration in Aging, and by the Department of Medicine Program of Experimental Medicine Research Award, the University of Western Ontario. The funders involved in setting up and maintaining the ADNI national database, where we obtained our data, are listed in the Acknowledgements. Neither ADNI nor its funders, some of which were commercial, have had any communication with us related to or influencing the specifics of our project. Their funding of ADNI does not alter our adherence to PLOS ONE policies on sharing data and materials.

**Competing interests:** No authors have competing interests.

## Conclusion

Adding motor function assessments to the ADAS-Cog may improve responsiveness in pre-dementia populations

## Introduction

The Alzheimer's Disease (AD) Assessment Scale–Cognitive Subscale (ADAS-Cog) was developed in 1984 to assess cognitive dysfunction in AD [1]. The ADAS-Cog subsequently became a widely adopted outcome measure for assessing efficacy in clinical trials of antidementia treatments and is still used today. Multiple studies have reported relationships between cognitive and motor function in pre-dementia syndromes [2–7]. There is a need for outcome measures that reflect these advancements and are more responsive for present research settings, while maintaining compatibility with historical measurement techniques.

Pre-dementia syndromes such as mild cognitive impairment (MCI) involve decreased cognitive functioning of memory, language, and judgement, with decrements in between normal or expected age-associated cognitive decline and serious cognitive and functional deficits seen with dementia [8,9]. There are concerns about the responsiveness of the ADAS-Cog at pre-dementia stages of disease [10–13]. Responsiveness is a form of validity defined as the ability to detect change [14–17]. Change can be contextualized using three aspects: group versus individual level of measurement, between-person versus within-person comparison, and the type of change one is interested in detecting [14]. The responsiveness of any outcome measure is population and context specific; although the ADAS-Cog performs well for studies of dementia it does not meet the needs of studies earlier in the natural history of disease progression [14,16].

Responsiveness can be affected by measurement properties such as floor and ceiling effects of individual items. Accordingly, several modifications have improved ADAS-Cog responsiveness, including alternative scoring, removing tasks, and adding assessments of delayed recall, executive function, and activities for daily living [10–12,18–23]. An important property when modifying an outcome measure is backward compatibility as this allows the direct comparison of novel study results with previous literature based on the original measure. An advantage of maintaining backwards compatibility with the ADAS-Cog is the ability to compare results with the vast amount of previously conducted literature that uses this 'de-facto' gold standard measure.

Since the time of ADAS-Cog development, research has found motor function decline plays an important role in dementia and pre-dementia syndromes [24,25]. Motor function tests help assess aspects of severity or stage of dementia not captured by purely cognitive tests [24,25]. Performance on tests such as quantitative gait assessment has been associated with cognitive status, changes in cognition over time, and incident dementia [1,26–34]. Furthermore, combined cognitive and gait impairments are more strongly associated with risk of cognitive decline and conversion to dementia than either component alone [31,35]. Dual-task gait performance (walking while simultaneously performing a cognitive task) has been associated with cognitive ability, different pre-dementia syndromes, and incident dementia [33,36,37]. The magnitude of change in gait during dual-tasking can be expressed as a dual-task gait cost (DTC), which adjusts for an individual's baseline gait characteristics [38]. Importantly, the ability to maintain gait control while using cognitive resources underlies the ability to safely perform daily activities required for independent living [39]. Our literature review of

modifications made to the ADAS-Cog since its development did not find any revisions whereby motor function or DTC assessments were added to the ADAS-Cog [13]. The overall aim of our study was to assess whether motor function assessments can be a helpful addition to cognitive outcome measures for detecting pre-dementia syndrome progression.

We hypothesized that adding assessments of motor function to the ADAS-Cog would improve responsiveness among older adults with pre-dementia syndromes. Our objectives were 1) to develop an outcome measure using a pooled index approach that includes quantitative gait and DTC assessments and is backwards-compatible to the ADAS-Cog and, 2) compare the responsiveness of the ADAS-Cog and the Pooled Index to group-level between-person differences in stage of pre-dementia disease progression at one point in time (baseline discrimination), and 3) compare the responsiveness of the ADAS-Cog, the Pooled Index, and different combinations of items to group-level within-person measured change over time in a pre-dementia sample (longitudinal decline).

## Methods

We searched for a database containing ADAS-Cog scores, quantitative gait assessments, and prospectively measured conversion to dementia across different cognitive subgroups at baseline. Because we did not locate a database with all required items, we accessed two partially overlapping datasets that together had the required variables and developed a statistical model that approximated the ADAS-Cog, the 'ADAS-Cog-Proxy'.

### Study populations

**The Gait and Brain Study.** The Gait and Brain Study is an ongoing prospective cohort (clinicaltrial.gov identifier: NCT03020381) designed to determine whether quantitative gait deficits can predict cognitive and mobility decline, falls, and progression to dementia among community-dwelling older adults. Study details have been described elsewhere [31,33,40]. The study was approved by the University of Western Ontario Health Sciences Research Ethics Board (approval number: 17200), and written informed consent was obtained from participants at the time of enrollment. Participant recruitment began in 2007 from geriatric and memory clinics at hospitals affiliated with the University of Western Ontario in London, Ontario. Inclusion criteria were 65 to 85 years old, able to walk 10 meters without assistance, and absence of dementia. Exclusion criteria were lack of English proficiency, Parkinsonism or other neurological disorder affecting motor function, musculoskeletal disorders or joint replacements that affect gait performance (clinician-assessed), use of psychotropics that can influence motor performance, and major depression. At baseline, eligible participants were divided into three diagnostic categories based on performance in cognitive testing and clinical evaluation. The Normal Cognition (NC) group had normal age-, sex-, and education-adjusted scores on the Mini-Mental State Examination (MMSE) [41] and Montreal Cognitive Assessment (MoCA) [42] based on standardized norms that account for age, sex, and education [43]. Subjective Cognitive Impairment (SCI) criteria were the same as that for NC, except patients reported persistent decline in cognition that was not explainable by an acute event, and answered yes to both, "Do you feel like your memory or thinking is becoming worse?" and "Does this concern you?". As described in the study protocol and following work [36,44], Mild Cognitive Impairment (MCI) was based on Petersen criteria [9] and included 1) a score of 0.5 on the Clinical Dementia Rating (CDR) Scale, 2) subjective cognitive complaints, 3) measured cognitive impairment in memory, executive function, attention, and/or language defined as scores 1.5 SD below expected performance based on published norms for age, sex, and education [43], 4) intact Lawton-Brody Activities of Daily Living, and 5) absence of dementia

determined by a specialized clinician and based on Diagnostic and Statistical Manual of Mental Disorders version IV-TR criteria.

**The Alzheimer's Disease Neuroimaging Initiative.** The Alzheimer's Disease Neuroimaging Initiative (ADNI) is a multi-phase study that began in 2003 as a public-private study partnership with the primary goal of testing whether neuroimaging, biological, and clinical assessments can be combined to measure progression from MCI to early AD (*adni.loni.usc.edu*). Study sites are located throughout North America. We used data from the first phase, ADNI1; detailed information on ADNI1 can be found at *www.adni-info.org*. Because ADNI1 collected the ADAS-Cog as well as several of the same cognitive measures administered in the Gait and Brain Study, ADNI1 data were used to develop a predictive model that estimates ADAS-Cog scores (ADAS-Cog-Proxy) that would have been collected in the Gait and Brain Study. ADNI1 data was downloaded on October 26, 2016. General inclusion criteria: Hachinski Score less than or equal to 4, age 55–90 years, stability of ADNI-permitted medications for 4 weeks, Geriatric Depression Scale under 6, study partner with 10 hours or more per week contact to accompany the participant to visits, visual and auditory acuity adequate for neuropsychological testing, good general health, sterile or two years past childbearing potential for women, willing and able to complete a three year imaging study, completed six grades of education or has good work history, fluent English or Spanish speaking ability, commitment to Neuroimaging and no medical contraindications to MRI, agrees to DNA for ApoE testing and banking and to blood and urine for biomarkers, and not enrolled in other trials or studies [45].

## Measures

**Cognition.** The process from ADAS-Cog-Proxy model development in ADNI1 to score estimation for participants in the Gait and Brain Study is described in detail in the S1 Appendix. Briefly, a predictive model developed with ADNI1 data was used to obtain ADAS-Cog-Proxy scores. The outcome was ADAS-Cog score; candidate covariates were cognitive tests present in both ADNI1 and the Gait and Brain Study (MMSE, Rey Auditory Verbal Learning Test, CDR- Sum of Boxes, Digit Span Forward, Digit Span backward, Trail Making Test A, and Trail Making Test B). Linear Regression and Generalized Additive Model (GAM) predictive models were considered [46,47]. Candidate models were constructed in a development subset of baseline ADNI1 data, using different combinations of candidate covariates. Preliminary accuracy was assessed in the development subset as the percentage of participants with a predicted ADAS-Cog-Proxy score within three points of their observed ADAS-Cog score. Three points is considered a clinically significant change [48]. The best candidate ADAS-Cog-Proxy predictive model was selected based on preliminary accuracy estimates and on similarity of the measurement domains captured by covariates to the ADAS-Cog. Final model accuracy was estimated in a separate testing subset of ADNI1 data.

To allow all participants in the Gait and Brain Study to have an ADAS-Cog-Proxy GAM score, Multivariate Imputation by Chained Equations (MICE) was used to impute missing GAM covariate values [49,50]. Five imputed datasets were created; the ADAS-Cog-Proxy GAM was applied to each, and the mean of the five estimated scores for each participant was taken as their final ADAS-Cog-Proxy GAM score. MICE and estimation of ADAS-Cog-Proxy GAM scores for participants in the Gait and Brain Study was repeated for baseline, 6, 12, 24, 36, and 48 month follow-up visits. Additional details are in Table B of the S1 Appendix.

**Motor function.** We evaluated quantitative gait performance using an electronic walkway system (GAITRite™) [51]; participants walk along the walkway and several gait measurements are taken. To avoid measuring acceleration and deceleration phases, start and end points were

marked one metre away from the boundaries of a six metre recording distance. Participants were asked to walk as they usually would, and average values across the recording distance were used. Four testing conditions were performed, including one single-task and three dual-task conditions (see below). The following spatio-temporal gait parameters were considered based on prominence and utility in previous research: velocity, stride time, step time, stride length, step length, double support time, swing time, stride width, stride velocity, and cadence. The coefficient of variation ($\frac{Standard\ Deviation(SD)}{Mean \times 100}$) allows direct comparison of variability across variables measured using different units, [52] and was calculated for all parameters except velocity.

**Motor-cognitive performance.** Simultaneous assessment of motor and cognitive functioning was done using the same electronic walkway as for motor function measurements, but participants were asked to perform cognitive tasks while walking. We assessed the three dual-task gait conditions of walking while: i) counting backwards from 100 by ones, ii) counting backwards from 100 by sevens, and iii) naming animals. Participants were not instructed to prioritize either the cognitive or the walking task. DTC was calculated for velocity, stride time, and stride time coefficient of variation, using the formula $\left[\frac{(Single-task\ condition-dual-task\ condition)}{single-task\ condition}\right] \times 100$ [36]. These three parameters were selected based on literature supporting their importance in dementia and pre-dementia syndromes [25,33,36,40].

**Baseline descriptive statistics.** Participant characteristics summarized for the two datasets included demographics (age, education, sex), medication count, comorbidity count, depressive symptoms (Geriatric Depression Scale [53], physical activity, Activities of Daily Living (Lawton-Brody Scale, Instrumental and Basic [54], Cognition [1] or ADAS-Cog-Proxy, MoCA [42], MMSE [41], CDR-SB [55], Rey Auditory Verbal Learning Test [56], Trail Making Test A and B [57], gait (velocity, stride time, strive time coefficient of variation [25,33,2]), and dual-task gait (gait velocity with counting, stride time with serial sevens, stride time with naming animals [25, 33, 2]).

## Analyses

**Pooled index development.** Our outcome measure was developed as a Pooled Index, which allows source variables with different scoring ranges to be combined into a single summary score while maintaining the ability to use each of the source variables individually [58–60]. Including up to six component variables with low pairwise correlations in a PI is recommended for covering important measurement domains and reducing variability of final PI scores [17–19]. We split our candidate components into three categories that have evidence of importance for pre-dementia and dementia syndromes and required at least one variable from each of the categories to be included in the Pooled Index: cognition, motor function, and motor-cognitive performance. After recoding variables so that higher values indicated greater dysfunction, Pooled Index scores were obtained by calculating Z-scores for each variable $\left[Z = \frac{(observation-goup\ mean)}{SD}\right]$, and then averaging those Z-scores [58,60]. When individual Pooled Index variables have low pairwise correlations, the SD of the combined score decreases as the number of source variables increases, which increases the detectable signal relative to noise or variability [59,60]. Variable selection for our Pooled Index was thus guided by low pairwise correlation coefficients, aided by theoretical considerations. An overview of our process is in Fig 1. To create the Pooled Index we first assessed pairwise correlations between the ADAS-Cog-Proxy GAM scores and each of the single-task gait and DTC variables. Variables were retained when |rho|<0.2 or when |rho| = 0.2 to 0.4 with evidence supporting that parameter's involvement in dementia or pre-dementia syndromes.

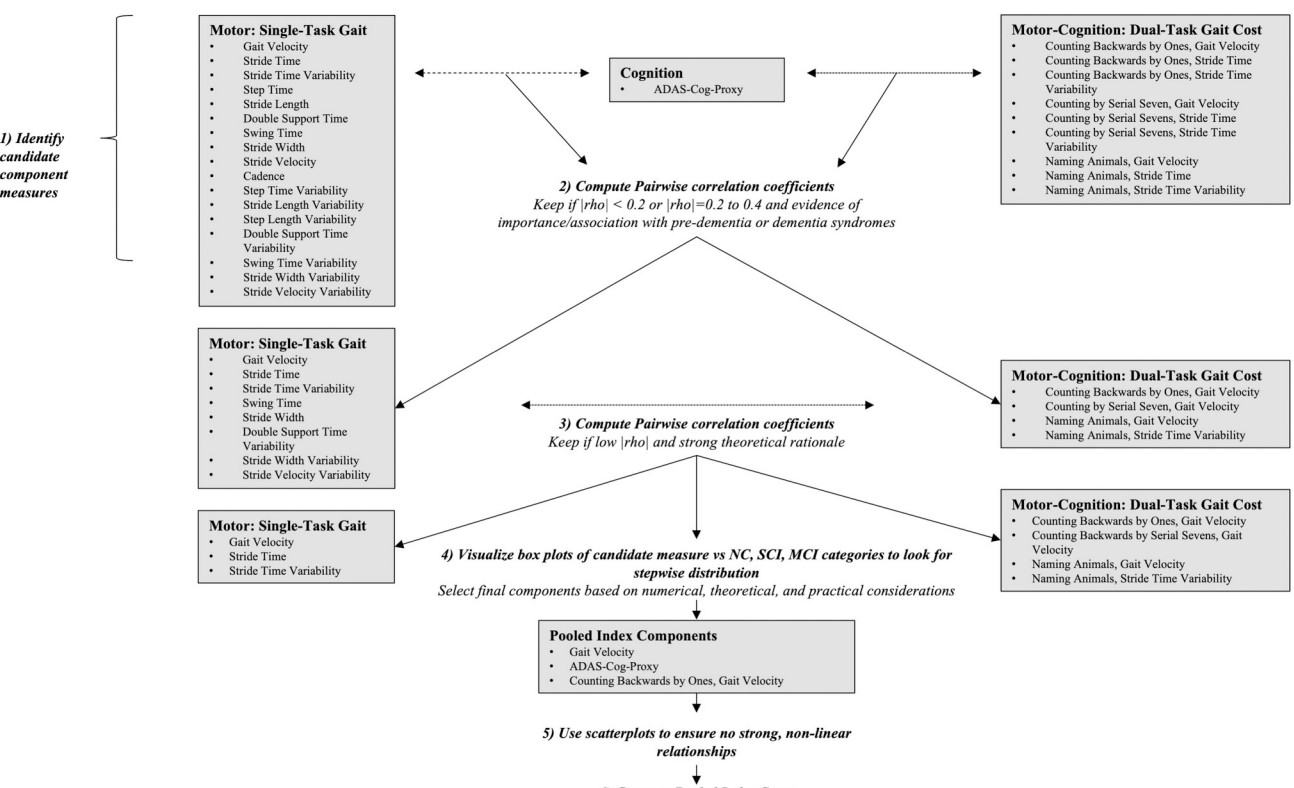

**Fig 1. Overview of pooled index development.** NC = Normal Cognition, SCI = Subjective Cognitive Impairment, MCI = Mild Cognitive Impairment.

Pairwise correlation coefficients were calculated for all retained single-task gait and DTC variables. In looking for at least one weakly intercorrelated pair, when numerical considerations were similar, we consulted literature on the involvement of the candidate variables in pre-dementia or dementia syndromes. When numerical and theoretical criteria were similar, box plots were created to assess which, if any, of the contending gait and DTC variables demonstrated a stepwise progression from NC to SCI to MCI diagnostic categories. Scatterplots were used to rule out strong non-linear relationships. Because most of the reduction in pooled SD occurs up to about six variables and diminishes thereafter, we focused our attention on finding a relatively small number of variables rather than the largest number possible. Finally, ease of assessment was a final pragmatic consideration for both individual variables and the Pooled Index as a whole.

**Responsiveness.** To assess whether motor function assessments improve responsiveness to changes in pre-dementia syndromes, we compared the Pooled Index to the ADAS-Cog-Proxy, which is standing in for the 'gold standard' original version of this cognitive outcome measure. Larger responsiveness test statistics suggest better detection of change.

Due to skewness and small sample sizes non-parametric tests were used to evaluate responsiveness to baseline discrimination. Kruskal-Wallis tests were used to assess whether the ADAS-Cog-Proxy and Pooled Index could detect a significant difference among the diagnostic categories of NC, SCI, and MCI. Mann-Whitney U tests were used to assess all pairwise comparisons.

Standardized Response Means $\left[ SRM = \frac{mean\ difference\ score}{SD\ of\ the\ difference\ score} \right]$ were used to assess responsiveness to longitudinal change over 6 to 48 months of follow-up for the Pooled Index, the ADAS-Cog-Proxy, and the ADAS-Cog-Proxy combined with individual Pooled Index components. Standardization was always performed with respect to the baseline distribution of participants

present at the follow-up visit of interest. Ninety-five percent bootstrap confidence intervals based on 1000 iterations of sampling with replacement were computed for the ADAS-Cog-Proxy GAM scores and for the Pooled Index.

*Secondary analysis.* SRMs and 95% bootstrap confidence intervals were calculated for the MMSE as a secondary analyses and used as a final comparison metric.

All analyses were conducted with RStudio, version 1.0.136 [61].

## Results

A total of 109 participants from the Gait and Brain Study were used to develop our Pooled Index and assess responsiveness. Baseline Gait and Brain Study characteristics can be found in Table 1. One participant with SCI did not have single-task gait recorded at baseline and was omitted from analyses. Participants who converted to dementia were included for time points prior to their dementia diagnosis. Two participants converted by six months of follow-up, one by 12 months, four by 24 months, and one by 36 months.

To develop the ADAS-Cog-Proxy, 573 participants from ADNI 1 were used; baseline ADNI1 characteristics can be found in Table 2.

### ADAS-Cog-Proxy

The best candidate model was a GAM with three covariates: Rey Auditory Verbal Learning Test (RAVLT), the MMSE, and the CDR-Sum of Boxes. Cognitive domains assessed by these measures overlap with the ADAS-Cog; adding more less similar cognitive assessments did not meaningfully improve GAM performance. Model accuracy on the ADNI1 testing subset was 69% of participant scores predicted within three points and 88% within five points of observed ADAS-Cog scores. A summary of ADAS-Cog-Proxy GAM development is in the S1 Appendix.

A summary of the number of missing GAM covariates imputed using MICE is in Table B of the S1 Appendix.

### Gait and cognition Pooled Index

Variables selected for the Pooled Index included the ADAS-Cog-Proxy, gait velocity, and DTC for gait velocity with the secondary task of counting backwards from 100 by ones (Fig 1). Pairwise correlation coefficients ranged from 0.27 to 0.32.

### Baseline discrimination

Both the ADAS-Cog-Proxy and the Pooled Index showed an overall statistically significant difference in mean ranks across the three diagnostic categories (ADAS-Cog-Proxy: $H(2) = 24.13$; PI: $H(2) = 22.36$, both $P<0.001$). Statistically significant pairwise comparisons were found for SCI versus MCI (ADAS-Cog-Proxy: $U = 331$, $P = 0.0002$; PI: $U = 348$, $P = 0.0009$) and NC versus MCI (ADAS-Cog-Proxy: $U = 153$, $P = 0.0002$; PI: $U = 148$, $P = 0.0001$), but not NC versus SCI diagnostic categories (ADAS-Cog-Proxy: $U = 93$, $P = 0.41$; PI: $U = 75$, $P = 0.17$).

### Longitudinal change

All SRMs are in Table 3. In terms of point estimates, the full Pooled Index demonstrated better responsiveness than the ADAS-Cog-Proxy GAM scores for 6 and 48 months, but not 36 months of follow-up. For 12 and 24 months the ADAS-Cog-Proxy GAM scores detected overall improvement, while the Pooled Index detected almost no change. Adding only gait velocity to the ADAS-Cog-Proxy using a Pooled Index approach consistently increased responsiveness

**Table 1. Gait and Brain Study baseline characteristics.**

| Characteristic Mean (SD) Minimum, Maximum Number of missing values (if applicable) *unless otherwise specified* | Overall[a] | Normal Cognition[b] | Subjective Cognitive Impairment[c] | Mild Cognitive Impairment[d] |
|---|---|---|---|---|
| Age (years) | 74.22 (6.33) | 73.50 (4.58) | 70.00 (4.59) | 75.36 (6.52) |
| | 63.00, 92.00 | 67.00, 82.00 | 65.00, 85.00 | 63.00, 92.00 |
| Education (years) | 13.85 (2.92) | 16.33 (3.06) | 14.42 (2.81) | 13.33 (2.74) |
| | 6.00, 20.00 | 10.00, 20.00 | 10.00, 20.00 | 6.00, 20.00 |
| Sex | | | | |
| Female n (%) | 58 (53) | 7 (58) | 15 (79) | 36 (49) |
| Male | 51 | 5 | 4 | 42 |
| Medications (#) | 7.62 (4.52) | 6.42 (4.06) | 6.53 (5.26) | 8.06 (4.37) |
| | 0, 21 | 2, 16 | 0, 21 | 0, 21 |
| Comorbidities (#) | 6.06 (2.85) | 4.33 (1.44) | 4.79 (2.02) | 6.64 (2.98) |
| | 0, 13 | 2, 7 | 1, 8 | 0, 13 |
| Geriatric Depression Scale | 2.35 (2.14) | 1.60 (1.14) | 2.25 (1.89) | 2.40 (2.21) |
| | 0, 10 | 0, 3 | 1, 5 | 0, 10 |
| | 22 | 7 | 15 | 0 |
| General Physical Activity Level | | | | |
| Vigorous: n (%) | 63 (58) | 6 (50) | 13 (68) | 44 (56) |
| Moderate: n (%) | 29 (27) | 5 (42) | 4 (21) | 20 (26) |
| Seldom: n (%) | 16 (25) | 1 (8) | 2 (11) | 13 (17) |
| Missing: n | 1 | 0 | 0 | 1 |
| Instrumental Activities of Daily Living | 7.69 (0.94) | 8.00 (0.00) | 7.75 (0.50) | 7.67 (0.99) |
| | 2, 8 | 8, 8 | 7, 8 | 2, 8 |
| | 22 | 7 | 15 | 0 |
| Basic Activities of Daily Living | 0.42 (0.97) | 0.80 (0.84) | 0.75 (0.96) | 0.38 (0.98) |
| | 0, 5 | 0, 2 | 0, 2 | 0, 5 |
| | 22 | 7 | 15 | 0 |
| ADAS-Cog-Proxy | 9.46 (2.34) | 7.59 (1.32) | 7.96 (1.93) | 10.11 (2.24) |
| | 3, 16 | 4, 9 | 3, 12 | 5, 16 |
| Montreal Cognitive Assessment | 24.45 (3.82) | 27.25 (1.48) | 27.89 (2.45) | 23.18 (3.60) |
| | 12, 30 | 24, 30 | 21, 30 | 12, 30 |
| Mini-Mental State Examination | 27.74 (2.52) | 28.83 (1.80) | 28.89 (1.45) | 27.29 (2.69) |
| | 18, 30 | 24, 30 | 24, 30 | 18, 30 |
| Clinical Dementia Rating Scale | 0.99 (0.89) | 0.0 (0.0) | NA | 1.07 (0.88) |
| | 0.0, 4.0 | 0.0, 0.0 | NA | 0.0, 4.0 |
| | 68 | 9 | 19 | 40 |
| Rey Auditory Verbal Learning Test (3 trials) | 17.20 (5.35) | 23.40 (5.18) | 24.75 (6.65) | 16.34 (4.71) |
| | 8.0, 33.0 | 19.0, 32.0 | 17.0, 33.0 | 8.0, 28.0 |
| | 29 | 7 | 15 | 7 |
| Gait Velocity (cm/s) | 108.40 | 124.80 | 114.10 | 104.60 |
| | 21.27 | 15.78 | 17.59 | 21.47 |
| | 57.27, 165.2 | 99.65, 155.80 | 82.17, 141.00 | 57.27, 165.20 |
| | 1 | 0 | 0 | 0 |
| Stride Time (s) | 1.14 (0.10) | 1.11 (0.08) | 1.10 (0.08) | 1.16 (0.10) |
| | 0.93, 1.41 | 0.95, 1.20 | 0.97.0, 1.26 | 0.93, 1.41 |
| | 1 | 0 | 1 | 0 |

*(Continued)*

**Table 1.** (Continued)

| Characteristic Mean (SD) Minimum, Maximum Number of missing values (if applicable) *unless otherwise specified* | Overall[a] | Normal Cognition[b] | Subjective Cognitive Impairment[c] | Mild Cognitive Impairment[d] |
|---|---|---|---|---|
| Stride Time Coefficient of Variation (CV) (%) | 2.47 (1.48) | 2.08 (0.76) | 2.49 (2.02) | 2.53 (1.43) |
| | 0.62, 9.73 | 1.14, 4.04 | 1.16, 9.73 | 0.62, 7.89 |
| | 1 | 0 | 1 | 0 |
| Dual-Task Gait Velocity Cost with Counting (%) | 5.51 (10.68) | 3.10 (11.52) | 2.58 (5.48) | 6.55 (11.35) |
| | -16.04, 34.61 | -8.16, 34.61 | -11.05, 10.82 | -16.04, 31.12 |
| | 1 | 0 | 1 | 0 |
| Dual-Task Stride Time Cost with Serial Sevens (%) | -16.93 (18.42) | -24.06 (29.08) | -8.23 (9.87) | -17.86 (17.37) |
| | -75.93, 6.30 | -75.93, 3.74 | -38.54, 2.74 | -69.50, 6.30 |
| | 3 | 0 | 1 | 2 |
| Dual-Task Stride Time CV Cost with Naming Animals (%) | -133.40 (270.66) | -214.80 (416.11) | -44.54 (80.73) | -141.30 (269.59) |
| | -1382.00, 77.58 | -1382.0, 63.87 | -240.3, 53.55 | -1200.00, 77.58 |
| | 1 | 0 | 1 | 0 |

SD = Standard Deviation, NA = Not Applicable. Number of participants

a = 109

b = 12

c = 19

d = 78.

to decline (less negative or more positive SRM), while adding only DTC to the ADAS-Cog-Proxy showed mixed results. For each time point, the 95% bootstrap confidence intervals for the Pooled Index capture a higher range of SRMs than for the ADAS-Cog-Proxy GAM scores; however, for each timepoint the intervals overlap. Secondary analysis of the MMSE show that it had the highest SRM point estimate for all time points except six months (S1 Table).

Due to the nature of using data from an on-going cohort study, not all participants had a chance to reach all follow-up time points, which contributed to our small sample sizes especially at the longest points of follow-up. Additional reasons for not having a visit at all time points include conversion to dementia and drop out due to health conditions or death. We assessed baseline differences between participants who did versus did not have each follow-up visit. Participants with 24- and 48-month visits had statistically significantly faster gait speed than those who did not have those visits. There were no statistically significant differences in baseline gait velocity for the other lengths of follow-up, or for any follow-up length in age, education, DTC, or ADAS-Cog-Proxy scores.

## Discussion

This proof of principle study suggests a Pooled Index approach combining assessments of motor function with ADAS-Cog-Proxy GAM scores may have comparable or increased responsiveness to changes in pre-dementia syndromes compared to ADAS-Cog-Proxy GAM scores alone.

More specifically, the Pooled Index and the ADAS-Cog-Proxy demonstrated similar responsiveness to baseline discrimination. Both detected statistically significant differences between NC and MCI, and SCI and MCI, but not between NC and SCI categories. For all but one follow-up period the Pooled Index trended towards greater responsiveness to longitudinal

**Table 2. Alzheimer's Disease neuroimaging initiative baseline characteristics.**

| Characteristic | Mean (SD) Minimum, Maximum Number of missing values (if applicable) *unless otherwise specified* |
|---|---|
| Age (years) | 75.17 (6.56) |
| | 54.40, 89.60 |
| Education (years) | 15.84 (2.94) |
| | 6.00, 20.00 |
| Sex | |
| Female n (%) | 228 (40%) |
| Male | 345 |
| Alzheimer's Disease Assessment Scale-Cognitive Subscale | 9.51 (4.63) |
| | 0, 28 |
| Mini-Mental State Examination | 27.78 (1.84) |
| | 23, 30 |
| Rey Auditory Verbal Learning Test (3 trials) | 18.34 (5.64) |
| | 5, 38 |
| Clinical Dementia Rating Scale-Sum of Boxes | 1.02 (1.03) |
| | 0.00, 4.50 |
| Trail Making Test A | 41.41 (20.08) |
| | 17.00, 188.00 |
| | 4 |
| Trail Making Test B | 114.8 (65.62) |
| | 34.0, 348.0 |
| | 5 |
| Digit Span Forward Test | 6.64 (1.05) |
| | 4, 8 |
| Digit Span Backward Test | 4.71 (1.19) |
| | 0, 7 |

Number of participants = 573, all from ADNI1. Five ADNI 1 participants were missing at least one covariate value for the fifth ADAS-Cog-Proxy candidate model and were excluded solely from analyses pertaining to that model.

decline than the ADAS-Cog-Proxy, but 95% bootstrap confidence intervals always overlapped. For two follow-up periods the point estimate for the ADAS-Cog-Proxy GAM scores detected improvement while the Pooled Index detected worsening; 95% bootstrap confidence intervals for both outcomes cross the point of no change so interpretation of these estimates must be done with caution. Estimates suggesting group-level improvement may be capturing the fact that the trajectory from NC to dementia is not linear, and not all participants are expected to progress to dementia. Motor function decline may not follow the same trajectory as cognitive decline, and has been found to occur in advance of cognitive decline and further disease progression [25,28–31]. An additional possibility to explain improvement on ADAS-Cog-Proxy SRMs while Pooled Index SRMs suggest worsening is practice effects or other inconsistency due to multiple versions of the RAVLT [62,63], which is one of the GAM covariates—this may artificially improve scores. Importantly, changes in cognitive or motor function abilities alone, or in the ability to engage in motor and cognitive tasks simultaneously, are all important aspects of functionality. Further research is needed to assess whether including gait assessments provide a more realistic assessment of changes in overall disease severity than purely cognitive measures.

**Table 3. Standardized response means to assess responsiveness to longitudinal change in the Gait and Brain Study.**

| n | Months of follow-up | ADASp (95% CI) | ADASp + GV | ADASp + DTC | ADASp + GV + DTC (95% CI) |
|---|---|---|---|---|---|
| 86 | 6 | 0.14 (-0.08, 0.34) | 0.17 | 0.18 | 0.23 (0.01, 0.47) |
| 73 | 12 | -0.05 (-0.31, 0.17) | -0.02 | 0.03 | 0.06 (-0.18, 0.31) |
| 55 | 24 | -0.24 (-0.49, 0.03) | -0.11 | -0.07 | 0.01 (-0.27, 0.26) |
| 35 | 36 | 0.23 (-0.08, 0.55) | 0.34 | 0.11 | 0.18 (-0.15, 0.56) |
| 24 | 48 | 0.60 (0.22, 1.04) | 0.68 | 0.59 | 0.65 (0.31, 1.2) |

ADASp = Alzheimer's Disease Assessment Scale-Cognitive Subscale-Proxy, CI = bootstrap Confidence Interval, DTC = Dual Task Cost, GV = Gait Velocity, m = months, n = sample size, + indicates variables were combined using a pooled index approach.

The improvements in longitudinal change responsiveness demonstrated by the Pooled Index were made without including explicit tests of delayed recall or executive function. These cognitive abilities decline in pre-dementia syndromes but are not included on the original ADAS-Cog [1,12]; previous ADAS-Cog modifications incorporating them have found improvements to responsiveness in pre-dementia studies [11,12,19,20]. Given previously found associations between gait velocity and DTC with cognitive abilities, especially executive function, we suspect part of the responsiveness of the Pooled Index is due to gait assessments capturing changes in executive function in addition to motor function aspects of disease progression; assessments of motor and cognitive function are not mutually exclusive [39,64,65].

The results from this study are weaker than expected when viewed alongside the larger body of literature demonstrating associations of gait and dual-task cost with cognitive decline and dementia [25–30,33,34]. Within the last decade, a new predementia syndrome, motoric cognitive risk syndrome, including both cognitive and motor deficits was introduced; it has been found to be prevalent internationally and have an association with conversion to dementia, suggesting the relationship between cognitive and motor decline is widespread [35,66]. In terms of baseline discrimination, gait and dual-task parameters similar to those used in the present study have been found to distinguish between subtypes of MCI (amnestic vs. non-amnestic) and dementia [33,34,36]. In terms of longitudinal change, motor function, assessed with gait and dual task cost, has been found to occur in advance of cognitive decline and to predict future cognitive decline and conversion to dementia [1,26–30,32,37]. Further, the combination of gait and cognitive measures has been found to better predicted dementia than either test alone [31]. Despite these advancements in understanding the natural history of cognitive and motor decline there is less research on developing an outcome measure that aligns with these advancements and optimizes different types of responsiveness; the present study provides a starting point—given the findings are not as clear as anticipated this further highlights a need for further research.

Outcome measures that assess motor and cognitive abilities at the same time may reduce inefficiencies in testing protocols and better detect meaningful changes in functionality or overall disease severity. Further practical advantages of using quantitative gait assessments for outcome measurement include language independence, non-invasive administration procedures, measurement precision, and for the DTC paradigm each participant serves as their own control. Advantages of gait velocity specifically are that it can be easily measured using only a stopwatch.

Similar advantages of using gait measures in a research setting apply to a clinical setting where ease and comprehensiveness of measurement are a priority. For example, gait velocity is a marker of overall health and the amount of dementia pathology, e.g. beta amyloid in the brain, is not necessarily associated with cognitive or functional decline. Adding gait measures

to neuropsychological testing in a clinical setting is both feasible and may provide a more accurate picture of progression to dementia and of overall health more generally. To get to this point more research is needed to develop a valid and reliable measure that includes both cognitive and motor assessments and is associated with clinically relevant outcomes. Testing of the measure would need to happen to assess responsiveness in both research and clinical settings as good performance in one setting does not always transfer to another [14–16].

An additional contribution of this study is our framework for developing the ADAS-Cog-Proxy. The process outlined in the S1 Appendix may be followed when there is an appropriate research question but not all necessary variables present in a single available database. Using a predictive model to obtain estimates of a missing variable allows preliminary tests of hypotheses without the time and resources that would be required to collect new data.

Main limitations of our study include small sample sizes, which may have contributed to some of the inconsistency in responsiveness across time points, missing data, and not using the original ADAS-Cog. The large proportion of missing data for the RAVLT and CDR-SB especially at longer follow-up visits may reduce the validity of imputation. There is a possible censoring bias given not all participants reached all time points, which is further suggested by the time points where participants who have the visit had faster baseline gait velocity (associated with overall health) than those who did not reach that time point. Our results should be replicated when a dataset with both ADAS-Cog and gait parameters collected under a common protocol becomes available. Two ADAS-Cog-Proxy GAM covariates were collected one month prior to ADAS-Cog administration, which may have contributed extra noise to the GAM development and led to an underestimate of accuracy. Restricting our Pooled Index to only gait velocity single and DTC with the ADAS-Cog-Proxy represents the trade-off in information value between practicality and measurement intensiveness. The derived units of the Pooled Index are also difficult to interpret and not directly comparable to ADAS-Cog scores.

In conclusion, our study used a proof of principle approach to explore whether adding motor tests to the ADAS-Cog would increase responsiveness to cognitive status and longitudinal changes. Our findings indicate a need for future research and researchers who are planning pre-dementia studies or developing new outcome measures may consider including gait assessments as part of a comprehensive test battery. Future steps include replicating the Pooled Index using the original ADAS-Cog, assessing responsiveness with larger subsamples of converters across all levels of disease severity from NC to dementia, further investigating direction of change identified by motor and cognitive measures, and assessing responsiveness to treatment effects in pre-dementia populations.

## Supporting information

**S1 Appendix. ADAS-Cog-Proxy development details.**
(DOCX)

**S1 Table. Secondary analysis: Mini mental state examination responsiveness to group-level within-person measured change over time.**
(DOCX)

## Acknowledgments

Portions of this study used data from the Alzheimer's Disease Neuroimaging Initiative (ADNI). Thus, data collection and sharing for this project was partially funded by ADNI (National Institutes of Health Grant U01 AG024904) and DOD ADNI (Department of Defense award number W81XWH-12-2-0012). ADNI is funded by the National Institute of

Biomedical Imaging and Bioengineering, and through contributions from: AbbVie, Alzheimer's Association; Alzheimer's Drug Discovery Foundation; Araclon Biotech; BioClinica, Inc.; Biogen; Bristol-Myers Squibb Company; CereSpir, Inc.; Cogstate; Eisai Inc,; Elan Pharmaceuticals, Inc.; Eli Lilly and Company; Eurolmmun; F. Hoffmann-La Roche Ltd and its affiliated company Genentech, Inc.; Fujirebio; GE Healthcare; IXICO Ltd.; Janssen Alzheimer Immunotherapy Research & Development, LLC.; Johnson & Johnson Pharmaceutical Research & Development LLC.; Lumosity; Lundbeck; Merk & Co., Inc.; Meso Scale Diagnostics, LLC.; NeuroRx Research; Neurotrack Technologies; Novartis Pharmaceuticals Corporation; Pfizer Inc.; Piramal Imaging; Servier; Takeda Pharmaceutical Company; and Transition Therapeutics. The Canadian Institutes of Health Research is providing funds to support ADNI clinical sites in Canada. Private sector contributions are facilitated by the Foundation for the National Institutes of Health (www.fnih.org). The grantee organization is the Northern California Institute for Research and Education, and the study is coordinated by the Alzheimer's Therapeutic Research Institute at the University of Southern California. ADNI data are disseminated by the Laboratory for Neuro Imaging at the University of Southern California.

A listing of all ADNI investigators, their role with ADNI, and affiliation can be found at: http://adni.loni.usc.edu/wp-content/uploads/how_to_apply/ADNI_Acknowledgement_List. pdf The principal investigator is Michael W. Weiner, MD and e-mail addresses categorized by type of question can be found at: http://adni.loni.usc.edu/about/contact-us/

## Author Contributions

**Conceptualization:** Jacqueline K. Kueper, Daniel J. Lizotte, Manuel Montero-Odasso, Mark Speechley.

**Formal analysis:** Jacqueline K. Kueper.

**Funding acquisition:** Jacqueline K. Kueper, Manuel Montero-Odasso, Mark Speechley.

**Methodology:** Jacqueline K. Kueper, Daniel J. Lizotte, Manuel Montero-Odasso, Mark Speechley.

**Resources:** Manuel Montero-Odasso, Mark Speechley.

**Supervision:** Daniel J. Lizotte, Manuel Montero-Odasso, Mark Speechley.

**Writing – original draft:** Jacqueline K. Kueper.

**Writing – review & editing:** Jacqueline K. Kueper, Daniel J. Lizotte, Manuel Montero-Odasso, Mark Speechley.

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
