## [Decision Letter · Decision Letter 0]

15 Nov 2019

PONE-D-19-25160

Cognition and Motor Function: The Gait and Cognition Pooled Index

PLOS ONE

Dear Dr. Montero-Odasso,

Thank you for submitting your manuscript to PLOS ONE. As you will see below, the four reviewers all had significant concerns and therefore, after careful consideration, we feel that although it has merit it does not fully meet PLOS ONE’s publication criteria as it currently stands. Therefore, we invite you to consider submitting a revised version of the manuscript should you feel able to address the points raised during the review process.

We would appreciate receiving your revised manuscript by Dec 30 2019 11:59PM. To enhance the reproducibility of your results, we recommend that if applicable you deposit your laboratory protocols in protocols.io, where a protocol can be assigned its own identifier (DOI) such that it can be cited independently in the future. For instructions see: http://journals.plos.org/plosone/s/submission-guidelines#loc-laboratory-protocols

We look forward to receiving your revised manuscript.

Kind regards,

Antony Bayer

Academic Editor

PLOS ONE

Journal Requirements:

3. We understand that you use data from ADNI in your study. Please update your Data Availability Statement to indicate how these data can be obtained.

4. Thank you for stating the following in the Financial Disclosure section:

This work was supported by the Early Research Award (PI MMO) and the Alzheimer Foundation of London and Middlesex Master’s Scholarship in Alzheimer-Related Research to JKK.  

The Gait and Brain Study is funded by an operating grant from the Canadian Institutes of Health and Research (CIHR, MOP 211220) and a CIHR Project Grant (PJT 153100).

Dr. Montero-Odasso’s program in “Gait and Brain Health” is supported by grants from the CIHR, the Ontario Ministry of Research and Innovation, the Ontario Neurodegenerative Diseases Research Initiative, the Canadian Consortium on Neurodegeneration in Aging, and by the Department of Medicine Program of Experimental Medicine Research Award, the University of Western Ontario.

Portions of this study used data from the Alzheimer’s Disease Neuroimaging Initiative (ADNI). Thus, data collection and sharing for this project was partially funded by ADNI (National Institutes of Health Grant U01 AG024904) and DOD ADNI (Department of Defense award number W81XWH-12-2-0012). ADNI is funded by the National Institute of Biomedical Imaging and Bioengineering, and through contributions from: AbbVie, Alzheimer’s Association; Alzheimer’s Drug Discovery Foundation; Araclon Biotech; BioClinica, Inc.; Biogen; Bristol-Myers Squibb Company; CereSpir, Inc.; Cogstate; Eisai Inc,; Elan Pharmaceuticals, Inc.; Eli Lilly and Company; Eurolmmun; F. Hoffmann-La Roche Ltd and its affiliated company Genentech, Inc.; Fujirebio; GE Healthcare; IXICO Ltd.; Janssen Alzheimer Immunotherapy Research & Development, LLC.; Johnson & Johnson Pharmaceutical Research & Development LLC.; Lumosity; Lundbeck; Merk & Co., Inc.; Meso Scale Diagnostics, LLC.; NeuroRx Research; Neurotrack Technologies; Novartis Pharmaceuticals Corporation; Pfizer Inc.; Piramal Imaging; Servier; Takeda Pharmaceutical Company; and Transition Therapeutics. The Canadian Institutes of Health Research is providing funds to support ADNI clinical sites in Canada. Private sector contributions are facilitated by the Foundation for the National Institutes of Health (www.fnih.org). The grantee organization is the Northern California Institute for Research and Education, and the study is coordinated by the Alzheimer’s Therapeutic Research Institute at the University of Southern California. ADNI data are disseminated by the Laboratory for Neuro Imaging at the University of Southern California.

We note that you received funding from a commercial source: AbbVie, Alzheimer’s Association; Alzheimer’s Drug Discovery Foundation; Araclon Biotech; BioClinica, Inc.; Biogen; Bristol-Myers Squibb Company; CereSpir, Inc.; Cogstate; Eisai Inc,; Elan Pharmaceuticals, Inc.; Eli Lilly and Company; Eurolmmun; F. Hoffmann-La Roche Ltd and its affiliated company Genentech, Inc.; Fujirebio; GE Healthcare; IXICO Ltd.; Janssen Alzheimer Immunotherapy Research & Development, LLC.; Johnson & Johnson Pharmaceutical Research & Development LLC.; Lumosity; Lundbeck; Merk & Co., Inc.; Meso Scale Diagnostics, LLC.; NeuroRx Research; Neurotrack Technologies; Novartis Pharmaceuticals Corporation; Pfizer Inc.; Piramal Imaging; Servier; Takeda Pharmaceutical Company; and Transition Therapeutics.

6.  We note that you have indicated that data from this study are available upon request. PLOS only allows data to be available upon request if there are legal or ethical restrictions on sharing data publicly. For information on unacceptable data access restrictions, please see http://journals.plos.org/plosone/s/data-availability#loc-unacceptable-data-access-restrictions.

7. One of the noted authors is a group or consortium: The Alzheimer’s Disease Neuroimaging Initiative

In addition to naming the author group, please list the individual authors and affiliations within this group in the acknowledgments section of your manuscript. Please also indicate clearly a lead author for this group along with a contact email address.

Reviewers' comments:

Reviewer's Responses to Questions

**Comments to the Author**

1. Is the manuscript technically sound, and do the data support the conclusions?

Reviewer #1: No

Reviewer #2: Partly

Reviewer #3: Yes

Reviewer #4: Yes

2. Has the statistical analysis been performed appropriately and rigorously? 

Reviewer #1: Yes

Reviewer #2: Yes

Reviewer #3: Yes

Reviewer #4: Yes

3. Have the authors made all data underlying the findings in their manuscript fully available?

Reviewer #1: Yes

Reviewer #2: No

Reviewer #3: Yes

Reviewer #4: No

4. Is the manuscript presented in an intelligible fashion and written in standard English?

Reviewer #1: Yes

Reviewer #2: Yes

Reviewer #3: Yes

Reviewer #4: Yes

5. Review Comments to the Author

Reviewer #1: The study tried to increase the responsiveness of clinical assessments of dementia through combining cognitive assessment and gait assessments. The idea is very interesting, however, the hypothesis and methodology have drawbacks.

Major points

1) The responsiveness of the measurement is off course important, however, it should also reflect some pathological changes, not physiological aging process. Therefore, in this case the measurement should be sensitive to conversion to AD or worse outcomes, not general changes over time. However, the methodology of the current study did not consider pathological changes at all.

2) The authors used new cognitive index of ADAS-Cog-Proxy instead of well-established ADAS-Cog, which has no established evidence of accuracy nor responsiveness.

Minor point

1) Table 1 and 2 seems to be exchanged.

Reviewer #2: Thank you for the opportunity to review the manuscript ‘Cognition and Motor Function: The Gait and Cognition Pooled Index by Keuper et al.

This study set out to investigate if the addition of motor function assessment to a cognitive test, the ADAS-Cog, could increase detection of cognitive deterioration in older adults, with a proof of principle approach. Since a database with complete set of variables was not found, the study used one database (the ADNI) to develop a predictive model for estimating an ADAS-cog proxy, which was then applied to the second database (the GABS) after imputing missing with multiple imputation. The outcome measure was a Pooled Index, developed from combining the ADAS-cog proxy with a motor function test (gait velocity) and a motor-cognitive test (dual-task). The Pooled Index was compared with the ADAS-cog proxy for responsiveness, that is, if able to detect a difference between diagnostic categories (NC, SCI and MCI) in the GABS sample (n=109). Calculation of effects size used to assess responsiveness to longitudinal change over 6 to 48 months. The study concluded that the Pooled index had comparable or increased responsiveness to changes compared to the ADAS-cog proxy. The concept of this study was interesting, and potential implications for the evaluation of effects from interventions preventing or delaying incident dementia important. The method was innovative, however, in my opinion, major revision is needed to improve description of the method and statistical analyses, and appropriate conclusions supported by the findings, which are tenuous at best.

Please find below major and minor comments.

Major:

Introduction

1. The rationale for using ADAD-cog could be better explained in the introduction, to support the development of a proxy of the test. Why not use MMSE for example, since it was available in GABS?

2. In the introduction it is suggested that the recognition of the relation between motor function and cognition function has impacted the suitability of ADAS-cog. Is there evidence of it being unsuitable? Is it not only a method of increasing responsiveness of the test pre-dementia?

3. Line 86: How is the backwards-compatibility of the pooled index and the ADAS-cog analysed?

Method:

4. The development of the ADAS-cog proxy takes much space in the method considering the aim. Could perhaps parts of it be lifted into supplement to aid readability?

5. Line 113: References for MMSE, MOCA, CDR, Lawton, DSM-TRI-IV and V missing.

6. Line 119: What tests and cut-offs for ‘measured cognitive impairments in memory, executive, function, attention, and/or language were used?

7. Line 120: DSM-V criteria for diagnoses differs from DSM-TR-IV. Did it differ between studies? How was this handled?

8. Line 125: What were inclusion/exclusion criteria, and informed consent and ethics for ADNI? Were the two datasets comparable with regards to age, physical and cognitive status?

9. Line 153: In Fig 1 (stage 2) the preliminary accuracy of M4 was 65.6%, while M5 was 68.4%? Why was M4 chosen as the best candidate model and not M5? Similarities of covariates perhaps?

10. Line 155: The progression from M4 (MMSE, RAVLT, CDR-SB) to the ADAS-cog proxy GAM in ADNI is not clear. A consistent terminology, for example, the ADAS-cog proxy GAM, may help. Did the final GAM differ from M4?

11. Line 164: ‘missing GAM covariate values’ confusing, which actual variables does this term refer to?

12. Line 165: What covariates were used in the multiple imputation model to predict RAVLT, CDR-SB, CDR-SB&RAVLT?

13. Line 163: Considering that missing on the covariates RAVLT, CDR-SB, CDR-SB&RAVLT was generally over 50% at most follow-ups for CDR-SB, and as much as 80% at 48 months, why use only 5 imputed datasets? The large proportion of missing values may reduce the validity of the imputation, which could also be discussed, and mentioned as a limitation. Furthermore, are data available with regards to reasons participants had missing values, particularly the CDR-SB?

14. Line 225: Were individual pooled index components added for the purpose of sensitivity analysis?

Results:

15. Line 236: Table 1 (and Table 2). Values with 2 decimals appear to be used as standard, but does not make much sense for variables measured in years for example.

16. Line 244: Further description of baseline measurements, with references, in the methods section would aid understanding of variables included in Table 2 (and Table 1). For example, with which test was Instrumental ADLs, and Basic ADLs assessed?

17. Line 270: Is the differences in responsiveness between the ADAS-cog proxy and the Pooled index tested statistically? Larger responsiveness suggests better detection of change, but differences seem very small. May the differences be due to chance? Since this is one of the main aims of this study, and on which main inferences are based, it would strengthen inferences if the difference was tested statistically. What size of change could be considered important?

18. Line 283: Is not having reached longer follow-up time points yet (due to staggered inclusion assumingly) the only reason for loss of data in later follow-ups? How do you explain participants with the longest follow-ups having faster gait speed compared with participants that have not yet reached those follow-up lengths?

19. Line 285: Was this comparison at baseline?

Discussion:

20. It is mentioned in the introduction that other studies have combined cognitive and motor components. How does method and results compare?

21. Line 291: This is a very generous (and also perhaps a little misleading) interpretation of results from the study. May be improved by being more specific for results for baseline discrimination (comparable results) and for responsiveness to changes (suggested results).

22. Line 297: This interpretation seems different from that in results section?

23. Line 299: The ADASp being the golden standard, is it not a big problem that the Pooled index detected worsening when the ADASp is suggesting improvement?

24. Line 300: While gait deterioration may have a different trajectory to cognitive decline, why does the ADASp+GV (i.e. only gait velocity added) follow a similar pattern to the ADASp?

25. Line 325: Specify how including motor function was a valuable contribution?

26. Since the Pooled index and the ADAS-cog proxy showed comparable baseline discrimination, and differences between tests on responsiveness to longitudinal changes did not appear to be statistically analysed, the inferences from this study seem tenuous?

Minor:

Title page:

27. Line 3: What is the order of authors? The order on the title page differs compared to page 1.

Abstract:

28. This being a proof of principle study could be mentioned.

Introduction:

29. Line 71: Can the order of references be arranged in sequence?

30. Line 79: Does the reference #33 compare dual-task in relation to ADL activities?

Methods:

31. Line 129: Does ADNI and ADNI 1 refer to the same study? (see also Line 133, 134, 138, 237, 241).

32. Line 143: How was similarity of covariates assessed?

Discussion:

33. Line 321: Is this short paragraph intended to be part of the previous?

Reviewer #3: Using data from ADNI, this study developed a pooled index combining gait assessment and ADAS-Cog, and further compared the responsiveness of pooled index to ADAS-Cog-Proxy both cross-sectionally and longitudinally in Gait and Brain Study (GABS). Authors showed comparable responsiveness to baseline discrimination. The pooled index, adding motor function assessment to ADAS-Cog, improved the responsiveness at 6- and 48-month follow-up but not at 36-month follow-up.

The rationale and analytical approaches are sound. I have several comments.

Major:

1. Regarding motor measures examined, a stronger rationale is needed. This study included one single-task and three dual-task conditions and included mean performance and variability of spatiotemporal gait parameters. It may be helpful to have a summary lead sentence or a vocabulary of various gait measures.

2. Regarding longitudinal analyses, a potential censoring bias exists because not all were followed up for 48 months. Although authors briefly mentioned this limitation, a sensitivity analysis should be considered to confirm findings.

3. Authors observed that ADAS-Cog-Proxy detected improvement and discussed a potential non-linear change. I wonder whether this reflects a learning effect from the memory test since the proxy included RAVLT. Perhaps authors can test this using GABS data.

Minor:

Line 112-113: please specify “normal” scores on MMSE and MoCA.

Line 115-116: please specify “persistent decline in cognition”. Between two consecutive assessments? Which cognitive function measure?

Line 149: were samples randomly divided into the testing and development subsets?

Line 153: it is not clear how the best candidate model was determined. Step 3 indicates the best candidate model is M4 and its preliminary accuracy is not the highest.

Line 164: what’s the criteria for imputation? It would be helpful to indicate missing here in the text, <5% for instance.

Line 165: please clarify “five datasets”

Line 188: It would be helpful to demonstrate pooled index development as a figure. I think that Figure 1 is a good illustration to show how ADAS-Cog proxy was developed in GABS.

Line 242: Please clarify Table 2 shows sample characteristics in GABS. Perhaps add “GABS” prior to “baseline”.

Reviewer #4: This is a technical article describing a pooled index approach to improve responsiveness in the ADAS-Cog in pre-dementia populations. This article could be improved by making it more interesting and clinically relevant to clinicians as well as researchers. Portions of the abstract, Introduction and discussion should be revised to describe the problem. Why is it important to create a tool that is more responsive in individuals with pre-dementia syndrome? Why is it important to distinguish between SCI and MCI.

In the abstract and introduction the authors take a historical approach ("The shift in focus in dementia research......"). However, this "shift" is not new and has been evolving in the research for over 15 years.

The first few lines for the abstract should be revised to reflect our current state of the science and the need for a more responsive measures. Revising the approach and tone of the introduction section should also be considered.

The introduction references studies related to 'Shifted attention" in research related to pre-dementia. Instead, I recommend defining key terms/conditions related to pre-dementia and summarize seminal studies that have demonstrated relationships between cognitive and motor function.

The discussion would also benefit from a discussion of how this work relates to prior studies, such a Verghese motot cognitive syndrome.

Also, what needs to be done to make this a clinically useful tool? What is the clinical relevance of this study?

6. PLOS authors have the option to publish the peer review history of their article (what does this mean?). If published, this will include your full peer review and any attached files.

Reviewer #1: No

Reviewer #2: No

Reviewer #3: No

Reviewer #4: No

---

## [Author Response · Author response to Decision Letter 0]

17 Jun 2020

Journal Requirements:

Response: We have updated the formatting to meet these PLOS ONE style requirements. 

Response: We have updated Supporting Information files and their referencing. 

3. We understand that you use data from ADNI in your study. Please update your Data Availability Statement to indicate how these data can be obtained.

Response: We have revised the Data Availability section of the submission portal to have statements for both the Gait and Brain Study data and for ADNI. Since only one drop down menu option is allowed, we maintained ‘No-some restrictions apply’ as this is true for the Gait and Brain Study data (see below), however we added to the description box a link to the data access application page on ADNI’s website. (http://adni.loni.usc.edu/data-samples/access-data/)

4. Thank you for stating the following in the Financial Disclosure section:

This work was supported by the Early Research Award (PI MMO) and the Alzheimer Foundation of London and Middlesex Master’s Scholarship in Alzheimer-Related Research to JKK. 

The Gait and Brain Study is funded by an operating grant from the Canadian Institutes of Health and Research (CIHR, MOP 211220) and a CIHR Project Grant (PJT 153100).

Dr. Montero-Odasso’s program in “Gait and Brain Health” is supported by grants from the CIHR, the Ontario Ministry of Research and Innovation, the Ontario Neurodegenerative Diseases Research Initiative, the Canadian Consortium on Neurodegeneration in Aging, and by the Department of Medicine Program of Experimental Medicine Research Award, the University of Western Ontario.

Portions of this study used data from the Alzheimer’s Disease Neuroimaging Initiative (ADNI). Thus, data collection and sharing for this project was partially funded by ADNI (National Institutes of Health Grant U01 AG024904) and DOD ADNI (Department of Defense award number W81XWH-12-2-0012). ADNI is funded by the National Institute of Biomedical Imaging and Bioengineering, and through contributions from: AbbVie, Alzheimer’s Association; Alzheimer’s Drug Discovery Foundation; Araclon Biotech; BioClinica, Inc.; Biogen; Bristol-Myers Squibb Company; CereSpir, Inc.; Cogstate; Eisai Inc,; Elan Pharmaceuticals, Inc.; Eli Lilly and Company; Eurolmmun; F. Hoffmann-La Roche Ltd and its affiliated company Genentech, Inc.; Fujirebio; GE Healthcare; IXICO Ltd.; Janssen Alzheimer Immunotherapy Research & Development, LLC.; Johnson & Johnson Pharmaceutical Research & Development LLC.; Lumosity; Lundbeck; Merk & Co., Inc.; Meso Scale Diagnostics, LLC.; NeuroRx Research; Neurotrack Technologies; Novartis Pharmaceuticals Corporation; Pfizer Inc.; Piramal Imaging; Servier; Takeda Pharmaceutical Company; and Transition Therapeutics. The Canadian Institutes of Health Research is providing funds to support ADNI clinical sites in Canada. Private sector contributions are facilitated by the Foundation for the National Institutes of Health (www.fnih.org). The grantee organization is the Northern California Institute for Research and Education, and the study is coordinated by the Alzheimer’s Therapeutic Research Institute at the University of Southern California. ADNI data are disseminated by the Laboratory for Neuro Imaging at the University of Southern California.

We note that you received funding from a commercial source: AbbVie, Alzheimer’s Association; Alzheimer’s Drug Discovery Foundation; Araclon Biotech; BioClinica, Inc.; Biogen; Bristol-Myers Squibb Company; CereSpir, Inc.; Cogstate; Eisai Inc,; Elan Pharmaceuticals, Inc.; Eli Lilly and Company; Eurolmmun; F. Hoffmann-La Roche Ltd and its affiliated company Genentech, Inc.; Fujirebio; GE Healthcare; IXICO Ltd.; Janssen Alzheimer Immunotherapy Research & Development, LLC.; Johnson & Johnson Pharmaceutical Research & Development LLC.; Lumosity; Lundbeck; Merk & Co., Inc.; Meso Scale Diagnostics, LLC.; NeuroRx Research; Neurotrack Technologies; Novartis Pharmaceuticals Corporation; Pfizer Inc.; Piramal Imaging; Servier; Takeda Pharmaceutical Company; and Transition Therapeutics.

Response: These commercial sources of funding were to ADNI investigators and none of the authors on this manuscript are ADNI investigators. The authors on this manuscript do not have any direct connection to this funding.

Response: The ORCID ID has been updated. 

6. We note that you have indicated that data from this study are available upon request. PLOS only allows data to be available upon request if there are legal or ethical restrictions on sharing data publicly. For information on unacceptable data access restrictions, please see http://journals.plos.org/plosone/s/data-availability#loc-unacceptable-data-access-restrictions.

Response: We have added this information to the cover letter, submission portal, and are repeating it here for ease; the rationale for not publicly data availability is mainly 3 fold:

1-The Gait and Brain study is in ongoing study and there are several research questions than need to be responded by PI and co-PIs before data can be released and available for researchers.

2-The Gait and Brain study started in 2007 and the original protocol and funding did not considered resources for releasing data.

3-Since 2014 it has been considered by the Pi and co-Pis the data can be released by per researchers’ request when study is finished, expected by 2025. This is upon resources can be secured to make them available to other researchers.

7. One of the noted authors is a group or consortium: The Alzheimer’s Disease Neuroimaging Initiative

In addition to naming the author group, please list the individual authors and affiliations within this group in the acknowledgments section of your manuscript. Please also indicate clearly a lead author for this group along with a contact email address.

Response: ADNI maintains a list of investigators and their affiliations as well as a webpage with current e-mail addresses to contact ADNI dependent on the type of inquiry. Direct links to these locations have been added to an acknowledgements section; copying the information into our manuscript would occupy over five pages of space. 

Reviewers' comments:

Reviewer's Responses to Questions

Comments to the Author

1. Is the manuscript technically sound, and do the data support the conclusions?

Reviewer #1: No

Reviewer #2: Partly

Reviewer #3: Yes

Reviewer #4: Yes

2. Has the statistical analysis been performed appropriately and rigorously?

Reviewer #1: Yes

Reviewer #2: Yes

Reviewer #3: Yes

Reviewer #4: Yes

3. Have the authors made all data underlying the findings in their manuscript fully available?

Reviewer #1: Yes

Reviewer #2: No

Reviewer #3: Yes

Reviewer #4: No

4. Is the manuscript presented in an intelligible fashion and written in standard English?

Reviewer #1: Yes

Reviewer #2: Yes

Reviewer #3: Yes

Reviewer #4: Yes

5. Review Comments to the Author

Reviewer #1: The study tried to increase the responsiveness of clinical assessments of dementia through combining cognitive assessment and gait assessments. The idea is very interesting, however, the hypothesis and methodology have drawbacks.

Major points

1) The responsiveness of the measurement is off course important, however, it should also reflect some pathological changes, not physiological aging process. Therefore, in this case the measurement should be sensitive to conversion to AD or worse outcomes, not general changes over time. However, the methodology of the current study did not consider pathological changes at all.

Response: While our research questions hypotheses were centred around change over time at a pre-dementia stages, we understand the hesitation to draw strong conclusions without a ‘harder’ outcome, especially given the absence of original ADAS-Cog scores. To get an additional outcome that is well-understood we performed secondary analyses with original MMSE scores. These results were added to the supplementary material. 

2) The authors used new cognitive index of ADAS-Cog-Proxy instead of well-established ADAS-Cog, which has no established evidence of accuracy nor responsiveness.

Response: the ADAS-Cog-Proxy scores are intended to estimate ADAS-Cog scores for the purposes of this proof of principle study; psychometric properties of the ADAS-Cog-Proxy depend on the performance of our predictive model. We do not advocate substituting the ADAS-Cog-Proxy for the ADAS-Cog in future study protocols. Rather, using a predictive model was a way to gain a preliminary look at our hypothesis before investing resources in a new study and may assist with future study measure selection. Nonetheless, we understand the concern about not having an original well-established cognitive measure to compare results to and have added a secondary analysis that uses MMSE scores for all time point. Since this is a secondary analysis we added it to the Supplementary Material, but reference it in the methods and provide a statement of the overall findings in the result section of the main manuscript. 

Minor point

1) Table 1 and 2 seems to be exchanged.

Response: Thank you for catching this error. Table locations have been fixed. 

Reviewer #2: Thank you for the opportunity to review the manuscript ‘Cognition and Motor Function: The Gait and Cognition Pooled Index by Keuper et al.

This study set out to investigate if the addition of motor function assessment to a cognitive test, the ADAS-Cog, could increase detection of cognitive deterioration in older adults, with a proof of principle approach. Since a database with complete set of variables was not found, the study used one database (the ADNI) to develop a predictive model for estimating an ADAS-cog proxy, which was then applied to the second database (the GABS) after imputing missing with multiple imputation. The outcome measure was a Pooled Index, developed from combining the ADAS-cog proxy with a motor function test (gait velocity) and a motor-cognitive test (dual-task). The Pooled Index was compared with the ADAS-cog proxy for responsiveness, that is, if able to detect a difference between diagnostic categories (NC, SCI and MCI) in the GABS sample (n=109). Calculation of effects size used to assess responsiveness to longitudinal change over 6 to 48 months. The study concluded that the Pooled index had comparable or increased responsiveness to changes compared to the ADAS-cog proxy. The concept of this study was interesting, and potential implications for the evaluation of effects from interventions preventing or delaying incident dementia important. The method was innovative, however, in my opinion, major revision is needed to improve description of the method and statistical analyses, and appropriate conclusions supported by the findings, which are tenuous at best.

Please find below major and minor comments.

Major:

Introduction

1. The rationale for using ADAD-cog could be better explained in the introduction, to support the development of a proxy of the test. Why not use MMSE for example, since it was available in GABS?

Response: We have revised the introduction with increased explanations of why the ADAS-Cog is used. Examples include stating early on that it is still in use today and that using it maintains compatibility with a large body of previously conducted research. We also added a sentence about how our methodological approach can be an example for other researchers who want to explore similar questions in other contexts. 

2. In the introduction it is suggested that the recognition of the relation between motor function and cognition function has impacted the suitability of ADAS-cog. Is there evidence of it being unsuitable? Is it not only a method of increasing responsiveness of the test pre-dementia?

Response: This comment raises concerns about the framing around the ADAS-Cog; we have revised the introduction to be more precise. For example, the ADAS-Cog is still suitable for its original purpose in dementia populations; it is not suitable for where the focus of present research is, but given its strong history may be able to maintain an important place. 

3. Line 86: How is the backwards-compatibility of the pooled index and the ADAS-cog analysed?

Response: Backwards compatibility is the ability to obtain the original measure from a new version. In our case, this can be achieved by using only the data collected about the ADAS-Cog without the data collected for other components of the pooled index. We specified how this is possible by extending the first sentence of the Analyses section to include, “while maintaining the ability to use each of the source variables individually”. 

In revising the introduction, we further expanded upon backwards compatibility and why it is important with the ADAS-Cog. 

Method:

4. The development of the ADAS-cog proxy takes much space in the method considering the aim. Could perhaps parts of it be lifted into supplement to aid readability?

Response: we have moved the figure describing the ADAS-Cog-Proxy GAM to Supplementary material, including the longer description that was below it. 

5. Line 113: References for MMSE, MOCA, CDR, Lawton, DSM-TRI-IV and V missing. 

Response: References have been added for MMSE and MoCA, which are mentioned on Line 113, as well as a reference for the source that was used to define NC/SCI and that includes the standardized norms used as cut offs. In clarifying the MCI definition, the other tests are no longer mentioned individually and instead there is increased description references to the source of the definitions are provided. 

6. Line 119: What tests and cut-offs for ‘measured cognitive impairments in memory, executive, function, attention, and/or language were used? 

Response: As per the study protocol and explanation in the following papers (1,2) MCI was defined following Petersen criteria and scores 1.5 SD below expected performance based on norms for age, sex, and education published in (3)

1. Montero-Odasso M, Muir SW, Speechley M. Dual-task complexity affects gait in people with mild cognitive impairment: The interplay between gait variability, dual tasking, and risk of falls. Arch Phys Med Rehabil. 2012;93(2):293–9. 

2. Annweiler, C., Beauchet, O., Bartha, R. et al. Slow gait in MCI is associated with ventricular enlargement: results from the Gait and Brain Study. J Neural Transm 120, 1083–1092 (2013). https://doi-org.proxy1.lib.uwo.ca/10.1007/s00702-012-0926-4

3. Strauss, E., Sherman, E. M. S., Spreen, O., & Spreen, O. (2006). A compendium of neuropsychological tests: Administration, norms, and commentary. Third Edition. Oxford: Oxford University Press.

7. Line 120: DSM-V criteria for diagnoses differs from DSM-TR-IV. Did it differ between studies? How was this handled?

Response: Thank you for identifying this. The Gait and Brain Study began using the DSM-V in 2013; the data for the present study comes from before then and so all diagnoses are based on DSM-IV. We have made this amendment in the manuscript. 

8. Line 125: What were inclusion/exclusion criteria, and informed consent and ethics for ADNI? Were the two datasets comparable with regards to age, physical and cognitive status?

Response: We added a summary of eligibility criteria for ADNI and a direct reference to the study protocol manual in the ADNI methods subsection. Given that both ADNI and GABS contain older adults along the pre- dementia disease continuum from NC to MCI, their cognitive abilities are expected to be similar. To ensure this was the case the range of ADAS-Cog-Proxy GAM covariates were compared between the ADNI data used to build the GAM and observed GABS data. The table used to compare this has been added to Supplementary Material. 

9. Line 153: In Fig 1 (stage 2) the preliminary accuracy of M4 was 65.6%, while M5 was 68.4%? Why was M4 chosen as the best candidate model and not M5? Similarities of covariates perhaps? 

Response: Yes, M4 was chosen because it was a simpler model without much worse performance. A key aspect of this similarity is that it includes covariates that measure similar domains to the ADAS-Cog whereas M5 includes assessments that measure additional domains; the generalizability of M4 is expected to be greater than M5. Clarification of these points were added to the Supplementary Material.

10. Line 155: The progression from M4 (MMSE, RAVLT, CDR-SB) to the ADAS-cog proxy GAM in ADNI is not clear. A consistent terminology, for example, the ADAS-cog proxy GAM, may help. Did the final GAM differ from M4? 

Response: Terminology has been updated so that ADAS-Cog-Proxy GAM (scores) is used throughout the manuscript to indicate predictions of ADAS-Cog scores in GABs. 

11. Line 164: ‘missing GAM covariate values’ confusing, which actual variables does this term refer to? 

Response: This refers to all of the covariates for M4. A reference to Table 1 of the Supplementary material, which lists these variables and the amount of missingness, has been added to Line 164. 

12. Line 165: What covariates were used in the multiple imputation model to predict RAVLT, CDR-SB, CDR-SB&RAVLT?

MICE was not performed on the entire GABS database due to multicollinearity and computational restrictions and because Research suggests that there is little improvement in accuracy when imputation considers more than 15-25 predictors. In accordance with published guidelines, predictor matrices included all GAM covariates, predictors of the outcome ADAS-Cog scores, variables that include a lot of variance as roughly identified by correlation with the target variables to be imputed, and no variables that had a lot of missing values within the subgroup of people with missing RAVLT and CDR-SB scores. It has also been suggested to include variables related to non-response. The main reason the CDR-SB scores are missing is if no collaborator was present to report on behalf of the patient; however, there was not a variable in the dataset expected to provide indication of this. The final list of included covariates for MICE was: Baseline Diagnosis, MMSE, MoCA, MoCAMIS, MoCAEIS, MoCAVIS, MoCALIS, MoCAAIS, MoCAOIS, CDR, Trail A, Trail B, Digit Forward, Digit Backward, Letter Number, RAVLT, BNT, FAB, number of falls in past 6 months, IADL, RSEO (balance), Gait Velocity, and Gait Velocity while counting backwards by ones, from the time point of interest, as well as CDR and RAVLT scores from the previous visit (T6 to T48 visit imputations) or a future visit (baseline visit imputations). The MICE procedure was performed separately for each time point to allow for the exclusion of observations that were missing simply because the corresponding participants did not have the follow-up visit. We added these details to Supplementary material.

1. van Buuren S, Groothuis-Oudshoorn K. mice: Multivariate imputation by chained equations in R. J Stat Softw. 2011;45(3). doi:10.18637/jss.v045.i03. 

2. van Buuren S, Oudshoorn K. Flexible Multivariate Imputation by MICE. Netherlands Organization for Applied Science Research;1999. 

3. White IR, Royston P, Wood AM. Multiple imputation using chained equations: issues and guidance for practice. Stat Med. 2011;30(4):377-399. doi:10.1002/sim.4067. 

13. Line 163: Considering that missing on the covariates RAVLT, CDR-SB, CDR-SB&RAVLT was generally over 50% at most follow-ups for CDR-SB, and as much as 80% at 48 months, why use only 5 imputed datasets? The large proportion of missing values may reduce the validity of the imputation, which could also be discussed, and mentioned as a limitation. Furthermore, are data available with regards to reasons participants had missing values, particularly the CDR-SB? 

Response: Numbers of missing values for the CDR-SB due to absence of a collaborator at appointments to fill out the assessment are available (the majority of cases) and have been added to Table 1 in Supplementary material. Reasons for missing RAVLT data are not available. High levels of missingness has been added as a limitation in the Discussion section. 

14. Line 225: Were individual pooled index components added for the purpose of sensitivity analysis? 

Response: analyses with sub-components of the pooled index were added to gain better insight into what types of change component measurement domains could capture and whether the trend across domains was similar. 

Results:

15. Line 236: Table 1 (and Table 2). Values with 2 decimals appear to be used as standard, but does not make much sense for variables measured in years for example. 

Response: decimals were included for variables measured in years as changes in cognition and motor function may occur on a monthly scale. We decided to keep it in in order to maintain a higher level of detail for comparison between subgroups. 

16. Line 244: Further description of baseline measurements, with references, in the methods section would aid understanding of variables included in Table 2 (and Table 1). For example, with which test was Instrumental ADLs, and Basic ADLs assessed? 

Response: Thank you for this suggestion to improve details. We added a “Baseline Descriptive Statistics” subsection to the Measures part of the methods that provides an overview of the baseline measurements with references. 

17. Line 270: Is the differences in responsiveness between the ADAS-cog proxy and the Pooled index tested statistically? Larger responsiveness suggests better detection of change, but differences seem very small. May the differences be due to chance? Since this is one of the main aims of this study, and on which main inferences are based, it would strengthen inferences if the difference was tested statistically. What size of change could be considered important? 

Response: We agree with you that additional information would be beneficial to better interpretation of results and computed bootstrap confidence intervals for the ADAS-Cog-Proxy GAM scores and for the Pooled Index at each time point, as well as for the secondary analysis of the MMSE outcome. Corresponding methods and findings have been incorporated into the manuscript, including tempering the final conclusions. 

18. Line 283: Is not having reached longer follow-up time points yet (due to staggered inclusion assumingly) the only reason for loss of data in later follow-ups? How do you explain participants with the longest follow-ups having faster gait speed compared with participants that have not yet reached those follow-up lengths? 

Response: Reasons for not reaching longer follow-up time points includes not being in the study long enough to reach all time-points, conversion to dementia, and drop out due to health conditions or death. These additional reasons have been added around Line 283. 

it is known that gait velocity is a good global marker of overall health so a possible explanation for participants with the longest follow-ups having faster gait speed compared with participants who have not yet reached those follow-up lengths is that those with higher gait speed at baseline have better health and are less likely to develop dementia (at which point they are taken out of the GABS) or drop out due to health conditions or death; they are more likely to remain healthy enough to stay in the study until longer follow-up points. 

19. Line 285: Was this comparison at baseline? 

Response: yes; we have added ‘baseline’ as a descriptor to the sentence to clarify this. 

Discussion:

20. It is mentioned in the introduction that other studies have combined cognitive and motor components. How does method and results compare? 

Response: Thank you for this suggestion to use our discussion section to compare our results with the broader literature base. We have added a paragraph to the discussion that describes in more detail some of the findings from key studies cited in the introduction that motivated the present study. 

21. Line 291: This is a very generous (and also perhaps a little misleading) interpretation of results from the study. May be improved by being more specific for results for baseline discrimination (comparable results) and for responsiveness to changes (suggested results). 

Response: We have updated this sentence to mention this is a proof of principle study with non-conclusive results. 

22. Line 297: This interpretation seems different from that in results section? 

Response: We double checked the discussion points with the results section and are not sure where the discrepancy is. We have removed “which were based on a primarily cognitive conceptualization of the disease” because in re-reviewing this may be a source of confusion.

23. Line 299: The ADASp being the golden standard, is it not a big problem that the Pooled index detected worsening when the ADASp is suggesting improvement? 

Response: This is a valid concern that highlights a ‘chicken-and-egg’ type of challenge. The ADASp became a ‘gold standard’ based on its performance in dementia trials; this status has been carried into other research study including observational and pre-dementia studies where it does not perform optimally. It additionally does not account for the now well-known motor components that accompany cognitive decline. So, rather than viewing this as a ‘problem’ we view it as a finding that needs to be further investigated. In conjunction with your point 20 suggestion, we have added to the discussion that our findings are not as clear as we expected and that more research is needed. Additionally, the now included 95% bootstrap confidence intervals show that in instances where the direction of change was opposite, the CI crosses the point of no change; the findings may be due to error and future research is needed to better understand this. 

24. Line 300: While gait deterioration may have a different trajectory to cognitive decline, why does the ADASp+GV (i.e. only gait velocity added) follow a similar pattern to the ADASp? 

Response: This is an excellent point which requires further research to untangle. We have added to our conclusions a future direction step to further investigate direction of change by motor and cognitive measures. 

25. Line 325: Specify how including motor function was a valuable contribution? 

Response: the contribution of investigating the including motor function is addressed throughout the manuscript as it is now understood to be associated with cognitive decline and dementia and there is a need for more responsive measures that align with this conceptualization. The paragraph beginning on Line 325 is focused on our predictive model work; the topic sentence has been updated to not mention motor function and avoid confusion. 

26. Since the Pooled index and the ADAS-cog proxy showed comparable baseline discrimination, and differences between tests on responsiveness to longitudinal changes did not appear to be statistically analysed, the inferences from this study seem tenuous? 

Response: to better understand the response to longitudinal changes 95% bootstrap confidence have been added for the ADAS-Cog-Proxy GAM scores and the Pooled Index at each time point. Throughout the manuscript we have tried to increase emphasis that this as a proof of principle study and to temper conclusions. 

Minor:

Title page:

27. Line 3: What is the order of authors? The order on the title page differs compared to page 1. 

Response: Thank you for identifying this inconsistency. The order on the title page is the correct one. We have updated the PlosOne information. 

Abstract:

28. This being a proof of principle study could be mentioned. 

Response: We have modified the descriptor in line three of the abstract from ‘pilot study’ to be ‘proof of principle study’.

Introduction:

29. Line 71: Can the order of references be arranged in sequence? 

Response: Thank you for identifying this discrepancy; we will fix the reference order. 

30. Line 79: Does the reference #33 compare dual-task in relation to ADL activities?

Response: Reference 33 investigates walking, tapping, and catching. Part of their hypothesis is that walking uses cognitive resources and will be more strongly correlated with the complex task of catching than the simpler/automatic task of tapping. 

Methods:

31. Line 129: Does ADNI and ADNI 1 refer to the same study? (see also Line 133, 134, 138, 237, 241). 

Response: ADNI is a multi-phase study; ADNI1 is the first phase. We have clarified this early in the ADNI section (line 123) and referred to ADNI as ADNI1 from thereon). 

32. Line 143: How was similarity of covariates assessed? 

Response: We have added that ‘similarity is based on measurement domains’. Additionally, the Supplementary Material now includes more details about the ADAS-Cog-Proxy model development including selection of the final model covariates. 

Discussion:

33. Line 321: Is this short paragraph intended to be part of the previous? 

Response: We intended for this short paragraph to stand on its own; we now moved the last sentence of the previous paragraph to this one. 

Reviewer #3: Using data from ADNI, this study developed a pooled index combining gait assessment and ADAS-Cog, and further compared the responsiveness of pooled index to ADAS-Cog-Proxy both cross-sectionally and longitudinally in Gait and Brain Study (GABS). Authors showed comparable responsiveness to baseline discrimination. The pooled index, adding motor function assessment to ADAS-Cog, improved the responsiveness at 6- and 48-month follow-up but not at 36-month follow-up.

The rationale and analytical approaches are sound. I have several comments.

Major:

1. Regarding motor measures examined, a stronger rationale is needed. This study included one single-task and three dual-task conditions and included mean performance and variability of spatiotemporal gait parameters. It may be helpful to have a summary lead sentence or a vocabulary of various gait measures. 

Response: Three categories to include in the PI were selected based on prior research suggesting importance for pre-dementia and dementia syndromes: cognition, motor function, and motor-cognitive performance. Including up to six component variables with low pairwise correlations in a PI is recommended for covering important measurement domains and reducing variability of final PI scores (9-11). So, we set out to include at least three variables from the aforementioned categories. All conditions assessed in GABS were considered; due to study restrictions and concerns about burden especially for an older population, conditions were selected based on previous research. Specific measures to use were selected based on presence and suggested importance in previous research; key studies are cited at the end of the DTC section. 

Additional information has been added to the motor, motor-cognitive performance, and pooled index development sections of the methods to increase rationale and description of the measurement procedures. We also added a reference to the GAITRite webpage where readers may find detailed information about the system and measurements. 

2. Regarding longitudinal analyses, a potential censoring bias exists because not all were followed up for 48 months. Although authors briefly mentioned this limitation, a sensitivity analysis should be considered to confirm findings.

Response: Thank you for this suggestion. We agree that there is potential bias that exists and have addressed this tangentially in another response—for some timepoints, those who remain in the study had faster gait velocity at baseline compared to those who did not and gait velocity is associated with overall better health. To help readers understand this we added additional reasons for not reaching a time point. We additionally added explicit mention of censoring bias to the limitation section. However, we decided not to perform a formal sensitivity analysis because that would require selecting expected outcomes for people who did not reach time points and it is unclear what the best or most unbiased method to do this would be, especially since we already know our primary results. 

3. Authors observed that ADAS-Cog-Proxy detected improvement and discussed a potential non-linear change. I wonder whether this reflects a learning effect from the memory test since the proxy included RAVLT. Perhaps authors can test this using GABS data. 

Response: this is in interesting observation and we found literature suggesting it is plausible. To avoid performing too many secondary analyses we have added this point to the discussion along with the references but did not test for this effect using GABS data. 

Minor:

Line 112-113: please specify “normal” scores on MMSE and MoCA. 

Response: we have added that ‘normal’ scores were based on standardized norms for age, sex, and education and provided a reference to the book these norms were taken from. 

Line 115-116: please specify “persistent decline in cognition”. Between two consecutive assessments? Which cognitive function measure? 

Response: this was a patient-reported outcome and not based on a cognitive function measure; SCI classification still required ‘normal’ scores on the MMSE and MoCA as defined for the NC group. 

Line 149: were samples randomly divided into the testing and development subsets?

Response: Yes; we added ‘randomly’ before the word divided. 

Line 153: it is not clear how the best candidate model was determined. Step 3 indicates the best candidate model is M4 and its preliminary accuracy is not the highest. 

Response: we have added additional details about how the best candidate model was determined to the supplementary material. 

Line 164: what’s the criteria for imputation? It would be helpful to indicate missing here in the text, <5% for instance. 

Response: All missing values were imputed. Additional details have been added to the Supplementary Material under Table S2, as requested by another reviewer. 

Line 165: please clarify “five datasets” 

Response: We are not sure what this comment is referring to; line 165 reads “Five imputed datasets”. 

Line 188: It would be helpful to demonstrate pooled index development as a figure. I think that Figure 1 is a good illustration to show how ADAS-Cog proxy was developed in GABS. 

Response: Thank you for this suggestion which we agree may provide additional helpful explanation to the Pooled Index development; we added a new figure. 

Line 242: Please clarify Table 2 shows sample characteristics in GABS. Perhaps add “GABS” prior to “baseline”. 

Response: We have clarified for Table 1 and 2, ‘GABS’ and ‘ADNI1’ characteristics. 

Reviewer #4: This is a technical article describing a pooled index approach to improve responsiveness in the ADAS-Cog in pre-dementia populations. This article could be improved by making it more interesting and clinically relevant to clinicians as well as researchers. Portions of the abstract, Introduction and discussion should be revised to describe the problem. Why is it important to create a tool that is more responsive in individuals with pre-dementia syndrome? Why is it important to distinguish between SCI and MCI.

Response: Thank you for this excellent suggestion to broaden the audience of our manuscript. Although the manuscript is targeted mainly at researchers we agree that adding functional or physical performance testing to global evaluation is of clinical relevance. It is known that the amount of dementia pathology, for example beta amyloid, in the brain is not necessarily associated with impeded cognition or functional performance. In fact. The strength of this association lessens with age. In other words, you may have a person in their 80s with lots of beta amyloid pathology without cognitive impairment. Therefore adding to neuropsychological testing functional or physical performance testing is expected to provide a more accurate picture of those with declining pathology and going to dementia. So, adding a physical test to assessments in a clinical setting may provide a better view of a patient’s health. 

Distinguishing between SCI and MCI was done to increase the rigour of testing for the Pooled Index—a measure that is capable of distinguishing NC, SCI, and MCI is expected to be able to detect more subtle changes than a measure that can only distinguish NC and MCI. It then becomes a choice for whoever is using the finalized measure about what differences are meaningful to detect. 

In the abstract and introduction the authors take a historical approach ("The shift in focus in dementia research......"). However, this "shift" is not new and has been evolving in the research for over 15 years.

The first few lines for the abstract should be revised to reflect our current state of the science and the need for a more responsive measures. Revising the approach and tone of the introduction section should also be considered.

Response: The abstract has been reframed to begin with a need for more responsive outcome measures, mention cognitive and motor decline, and state that this is a proof of principle study about the impact on responsiveness when motor function is added to a cognitive measure. The introduction has also been revised, as per your next point and suggestions from other reviewers. 

The introduction references studies related to 'Shifted attention" in research related to pre-dementia. Instead, I recommend defining key terms/conditions related to pre-dementia and summarize seminal studies that have demonstrated relationships between cognitive and motor function.

Response: We have revised the introduction, including an explanation of pre-dementia syndromes and removing the emphasis on ‘shifts’ to stating where the research is now and why we need research projects around motor and cognitive measurement. 

The discussion would also benefit from a discussion of how this work relates to prior studies, such a Verghese motot cognitive syndrome.

Response: This is a great idea and have added a paragraph to the discussion about prior studies, including MCRS. 

Also, what needs to be done to make this a clinically useful tool? What is the clinical relevance of this study?

Response: Continuing off of our response to your above suggestions about clinical relevance, we have added to the discussion a paragraph about clinical relevance of combining cognitive and motor measures and key check points before this type of measure would be ready for clinical use.

---

## [Decision Letter · Decision Letter 1]

18 Aug 2020

PONE-D-19-25160R1

Cognition and motor function: The gait and cognition pooled index

PLOS ONE

Dear Dr. Montero-Odasso,

Thank you for submitting your revised manuscript to PLOS ONE and for your detailed attention to the previous reviewers' comments. After careful consideration, we feel that it has considerable merit but does not fully meet PLOS ONE’s publication criteria as it currently stands. Therefore, we invite you to submit a revised version of the manuscript that addresses the few outstanding points raised during the latest review process.

We look forward to receiving your revised manuscript.

Kind regards,

Antony Bayer

Academic Editor

PLOS ONE

Reviewers' comments:

Reviewer's Responses to Questions

**Comments to the Author**

1. If the authors have adequately addressed your comments raised in a previous round of review and you feel that this manuscript is now acceptable for publication, you may indicate that here to bypass the “Comments to the Author” section, enter your conflict of interest statement in the “Confidential to Editor” section, and submit your "Accept" recommendation.

Reviewer #4: (No Response)

2. Is the manuscript technically sound, and do the data support the conclusions?

Reviewer #4: Yes

3. Has the statistical analysis been performed appropriately and rigorously? 

Reviewer #4: Yes

4. Have the authors made all data underlying the findings in their manuscript fully available?

Reviewer #4: Yes

5. Is the manuscript presented in an intelligible fashion and written in standard English?

Reviewer #4: Yes

6. Review Comments to the Author

Reviewer #4: The introduction and discussion of this paper are substantially improved. Recommended changes:

Abstract:

Line 36/37: incomplete sentence. Please revise.

Introduction:

Line 52: I disagree with the use of term “emerging research." Instead, “multiple studies have reported relationships between cognitive and motor function in pre-dementia syndromes". Citations are needed.

Here are just a few studies that could be cited:

Aggarwal NT, Wilson RS, Beck TL, Bienias JL, Bennett DA. Motor dysfunction in mild cognitive impairment and the risk of incident Alzheimer disease. Arch Neurol 2006;63:1763-9.

Mielke MM, Roberts RO, Savica R, et al. Assessing the Temporal Relationship Between Cognition and Gait: Slow Gait Predicts Cognitive Decline in the Mayo Clinic Study of Aging. J Gerontol A Biol Sci Med Sci 2013; 68:929-37.

Verghese J, Robbins M, Holtzer R, et al. Gait dysfunction in mild cognitive impairment syndromes. J Am Geriatr Soc 2008;56:1244-51

Buchman AS, Bennett DA. Loss of motor function in preclinical Alzheimer's disease. Expert Rev Neurother 2011;11:665-76.

Liu-Ambrose TY, Ashe MC, Graf P, et al. Increased risk of falling in older community-dwelling women with mild

cognitive impairment. Phys Ther. 2008; 88:1482–1491.

Waite LM, Grayson DA, Piguet O, et al. Gait slowing as a predictor of incident dementia: 6-year longitudinal data from the Sydney Older Persons Study. J Neurol Sci. 2005;229–230:89–93.

Line 53-55: The sentences "There is a need……” is vague. I recommend replacing with a clear and accessible statement.

Line 56: Mild Cognitive Impairment - No need to capitalize

Line 65: ….Natural history. add “of disease progression” or “of MCI” (Natural history of disease progression)

Line 87-88 Unclear sentence - “Our literature review found…

Line 91-93 Remove this statement: “ While we focus on the ADAS-Cog given its prominence and strong history in the field, but our methodological approach can serve as an example for researchers to follow for other contexts and measures as well.”

7. PLOS authors have the option to publish the peer review history of their article (what does this mean?). If published, this will include your full peer review and any attached files.

Reviewer #4: No

---

## [Author Response · Author response to Decision Letter 1]

20 Aug 2020

We are thankful to Reviewer 4 for going through our manuscript again and providing suggestions to improve its clarity. Please find our point by point responses below. 

Reviewer #4: The introduction and discussion of this paper are substantially improved. Recommended changes:

Abstract:

Line 36/37: incomplete sentence. Please revise.

Response: We have revised this sentence to: Final selected variables for the Pooled Index include gait velocity, dual-task cost of gait velocity, and an ADAS-Cog-Proxy (statistical approximation of the ADAS-Cog using similar cognitive tests).

Introduction:

Line 52: I disagree with the use of term “emerging research." Instead, “multiple studies have reported relationships between cognitive and motor function in pre-dementia syndromes". Citations are needed.

Here are just a few studies that could be cited:

Aggarwal NT, Wilson RS, Beck TL, Bienias JL, Bennett DA. Motor dysfunction in mild cognitive impairment and the risk of incident Alzheimer disease. Arch Neurol 2006;63:1763-9.

Mielke MM, Roberts RO, Savica R, et al. Assessing the Temporal Relationship Between Cognition and Gait: Slow Gait Predicts Cognitive Decline in the Mayo Clinic Study of Aging. J Gerontol A Biol Sci Med Sci 2013; 68:929-37.

Verghese J, Robbins M, Holtzer R, et al. Gait dysfunction in mild cognitive impairment syndromes. J Am Geriatr Soc 2008;56:1244-51

Buchman AS, Bennett DA. Loss of motor function in preclinical Alzheimer's disease. Expert Rev Neurother 2011;11:665-76.

Liu-Ambrose TY, Ashe MC, Graf P, et al. Increased risk of falling in older community-dwelling women with mild

cognitive impairment. Phys Ther. 2008; 88:1482–1491.

Waite LM, Grayson DA, Piguet O, et al. Gait slowing as a predictor of incident dementia: 6-year longitudinal data from the Sydney Older Persons Study. J Neurol Sci. 2005;229–230:89–93.

Response: We agree with the reviewer’s idea and are thankful for the suggested citations; these have been incorporated. 

Line 53-55: The sentences "There is a need……” is vague. I recommend replacing with a clear and accessible statement.

Response: We revised the sentence to read, “There is a need for outcome measures that reflect these advancements and are more responsive for present research settings, while maintaining compatibility with historical measurement techniques.”

Line 56: Mild Cognitive Impairment - No need to capitalize

Response: we removed the capitalization. 

Line 65: ….Natural history. add “of disease progression” or “of MCI” (Natural history of disease progression)

Response: we added ‘of disease progression’. 

Line 87-88 Unclear sentence - “Our literature review found…

Response: The sentence has been revised to: Our literature review of modifications made to the ADAS-Cog since its development did not find any revisions whereby motor function or DTC assessments were added to the ADAS-Cog

Line 91-93 Remove this statement: “ While we focus on the ADAS-Cog given its prominence and strong history in the field, but our methodological approach can serve as an example for researchers to follow for other contexts and measures as well.”

Response: We removed the sentence.

---

## [Editor Report · Decision Letter 2]

24 Aug 2020

Cognition and motor function: The gait and cognition pooled index

PONE-D-19-25160R2

Dear Dr. Montero-Odasso,

Thank you for you manuscript with further revisions. We’re pleased to inform you that this has been judged scientifically suitable for publication and will be formally accepted for publication once it meets all outstanding technical requirements.

Kind regards,

Antony Bayer

Academic Editor

PLOS ONE
---

## [Editor Report · Acceptance letter]

2 Sep 2020

PONE-D-19-25160R2 

Cognition and motor function: The gait and cognition pooled index 

Dear Dr. Montero-Odasso:

I'm pleased to inform you that your manuscript has been deemed suitable for publication in PLOS ONE. Congratulations! Your manuscript is now with our production department. 

Kind regards, 

on behalf of

Professor Antony Bayer 

Academic Editor

PLOS ONE